# Left Atrial Geometry and Phasic Function Determined by Cardiac Magnetic Resonance Are Independent Predictors for Outcome in Non-Ischaemic Dilated Cardiomyopathy

**DOI:** 10.3390/biomedicines9111653

**Published:** 2021-11-10

**Authors:** Bianca Olivia Cojan-Minzat, Alexandru Zlibut, Ioana Danuta Muresan, Rares-Ilie Orzan, Carmen Cionca, Dalma Horvat, Liliana David, Alexandru Ciprian Visan, Mira Florea, Lucia Agoston-Coldea

**Affiliations:** 1Department of Internal Medicine, Iuliu Hatieganu University of Medicine and Pharmacy, 400012 Cluj-Napoca, Romania; cojanminzat.bianca@yahoo.com (B.O.C.-M.); alex.zlibut@yahoo.com (A.Z.); ioanamuresandanuta@yahoo.com (I.D.M.); orzanrares@gmail.com (R.-I.O.); hdalma92@yahoo.com (D.H.); lilidavid2007@yahoo.com (L.D.); 2Department of Family Medicine, Iuliu Hatieganu University of Medicine and Pharmacy, 400012 Cluj-Napoca, Romania; miraflorea@yahoo.com; 3Department of Radiology, Affidea Hiperdia Diagnostic Imaging Center, 400012 Cluj-Napoca, Romania; carmen.cionca@yahoo.com; 42nd Department of Internal Medicine, County Emergency Hospital, 400012 Cluj-Napoca, Romania; 5Department of Cardiothoracic Surgery, Freeman Hospital, Newcastle-upon-Tyne NE7 7DN, UK; alexandruciprian.visan@gmail.com

**Keywords:** non-ischaemic dilated cardiomyopathy, cardiac magnetic resonance imaging, left atrial phasic function, left atrial volumes, left atrial sphericity index, left atrial strain

## Abstract

Left atrial (LA) geometry and phasic functions are frequently impaired in non-ischaemic dilated cardiomyopathy (NIDCM). Cardiac magnetic resonance (CMR) can accurately measure LA function and geometry parameters. We sought to investigate their prognostic role in patients with NIDCM. We prospectively examined 212 patients with NIDCM (49 ± 14.2-year-old; 73.5% males) and 106 healthy controls. LA volumes, phasic functions, geometry, and fibrosis were determined using CMR. A composite outcome (cardiac death, ventricular tachyarrhythmias, heart failure hospitalization) was ascertained over a median of 26 months. LA phasic functions, sphericity index (LASI) and late gadolinium enhancement (LA-LGE) were considerably impaired in the diseased group (*p* < 0.001) and significantly correlated with impaired LV function parameters (*p* < 0.0001). After multivariate analysis, LA volumes, LASI, LA total strain (LA-ε_t_) and LA-LGE were associated with increased risk of composite outcome (*p* < 0.001). Kaplan–Meier analysis showed significantly higher risk of composite endpoint for LA volumes (all *p* < 0.01), LASI > 0.725 (*p* < 0.003), and LA-ε_t_ < 30% (*p* < 0.0001). Stepwise Cox proportional-hazards models demonstrated a considerable incremental predictive value which resulted by adding LASI to LA-ε_t_ (Chi-square = 10.2, *p* < 0.001), and afterwards LA-LGE (Chi-Square = 15.8; *p* < 0.0001). NIDCM patients with defective LA volumes, LASI, LA-LGE and LA-ε_t_ had a higher risk for an outcome. LA-ε_t_, LASI and LA-LGE provided independent incremental predictive value for outcome.

## 1. Introduction

Non-ischemic dilated cardiomyopathy (NIDCM) is the commonest primary myocardial disorder, being characterized by systolic dysfunction due to left (LV) or bi-ventricular dilation, in the absence of coronary syndromes, arterial hypertension (AHT) or valvular heart disease (VHD) [1]. LA function is a key factor in maintaining LV haemodynamics. It is regularly impaired in various cardiac diseases and is strongly associated with mortality and poor outcomes [2,3]. Cardiac magnetic resonance imaging (CMR) is an advanced cardiovascular imaging technique that pledges to accurately characterize LA function and geometry, but studies which evaluate their predictive ability are still scarce, especially in NIDCM [4].

The interdependency between the LA and LV functions is widely recognized [5]. LA function is characterized by an unceasing physiological process which can be divided into three successive phases: LA reservoir function (LA filling in late LV systole), conduit function (passive LV filling in early diastole) and booster pump function (active LV filling in late diastole). Eventually, a LA suction force emerges which actively fills the LA in early LV systole [4,6]. Through this phasic phenomenon, LA has a central role in maintaining cardiac output even when the LV relaxation and compliance are defective [7]. In turn, the progression of LV diastolic dysfunction and increased filling pressures are key factors in promoting LA impairment and enlargement [5].

Late advances in the field of CMR have granted proper description of LA function and, thus, deploying specific parameters. These have comparable accuracy with speckle-tracking echocardiography in terms of LA strain and phasic function evaluation [8,9]. LA phasic functions were associated with the progression of heart failure (HF), LV diastolic dysfunction and increased filling pressures [10], and were independent predictors of major adverse cardiovascular events (MACEs) [11]. Moreover, LA strain parameters determined by CMR were able to distinguish patients with impaired LV relaxation from those without it [12]. However, their predictive roles in patients with NIDCM are still uncharted. 

In most cardiac conditions, altered LA geometry is strongly associated with cardiac dysfunction [13]. Maximum indexed LA volume (LAV_max_) is significantly correlated with MACEs, and current international guidelines recommend it as a marker of diastolic dysfunction [14,15], being also associated with increased mortality [16,17]. Moreover, LA sphericity index (LASI) is a novel marker that characterizes LA’s shape and remodelling which might play important roles in outcome prediction, but studies are just in their infancy [18]. LASI was able to predict the recurrence of atrial fibrillation [19] and HF hospitalization in NIDCM [20].

The aim of our study was to evaluate the role of LA geometry and function in the diagnosis and prognosis prediction in patients with NIDCM using CMR.

## 2. Materials and Methods

### 2.1. Study Design and Patients’ Characteristics

We conducted a prospective study on 212 patients with NIDCM and 106 healthy volunteers who were examined in the 2nd Department of Internal Medicine, Iuliu Hatieganu University of Medicine and Pharmacy from October 2017 to November 2020. The inclusion criteria comprised: (1) impaired LV ejection fraction (LVEF) ≤ 45%; (2) LV chamber dilation with indexed LV end-diastolic volume (LVEDV) ≥ 97 mL/m^2^; both being CMR-determined [1]. The exclusion criteria comprised: (1) ischemic heart disease which was defined as >50% angiographical stenosis in any epicardial coronary artery. Primary valvular disease was defined as moderate or higher valvular stenosis or regurgitation, apart from functional ones. Functional mitral regurgitation was defined as mitral regurgitation secondary to left ventricular remodeling resulting in failure of leaflet coaptation, in the setting of normal mitral valve anatomy, on echocardiography and CMR; (2) contraindications to CMR (incompatible metallic devices, significant chronic renal failure with estimated glomerular filtration rate < 30 mL/min/1.73 m^2^, or claustrophobia); (3) refusal to participate in the study (Figure 1). A total of 106 subjects free of overt cardiovascular disease, normal electrocardiogram, echocardiogram and negative CMR served as the control group.

Each patient underwent the same investigation protocol, including medical history, physical examination, 12-lead electrocardiogram, 24-h Holter monitoring, biochemical analysis, standard echocardiography and CMR.

### 2.2. CMR Imaging

All CMR images were ECG-gated and were acquired during apnoea with a 1.5 T Open Bore system MR scanner (Magnetom Altea, Siemens Medical Solutions, Erlangen, Germany). A standard scanning protocol in compliance with current international guidelines was used [21]. The acquisition of fast imaging employing steady-state free precession (SSFP) sequences was performed to detect ventricular function and mass in conventional cardiac short-axis and long-axis planes, to enclose both ventricles from base to apex. Sequence parameters SSFP were as follows: repetition time (TR) 3.6 ms; echo time (TE) 1.8 ms; flip angle 60°; slice thickness 6 mm; field of view 360 mm; image matrix of 192 × 192 pixels; voxel size 1.9 × 1.9 × 6 mm; 25–40 ms temporal resolution reconstructed to 25 cardiac phases. 

Late gadolinium enhancement (LGE) imaging, with sequence parameters of TR 4.8 ms, TE 1.3 ms, and inversion time 200–300 ms, was performed to detect myocardial fibrosis acquired 10 min after intravenous infusion of 0.2 mmol/kg gadoxetic acid (Clariscan, GH Healthcare AS, Oslo, Norway) in long- and short-axisviews, using a segmented inversion-recovery gradient-echo sequence. Inversion time was adjusted to optimize nulling of apparently normal myocardium. Brachial blood pressure was continuously monitored during SSFP-CMR acquisitions. 

Transmitral inflow and myocardial velocity ECG-gated phase-contrast CMR (PC-CMR) pulse sequences were used to acquire two series of images during two consecutive breath-holds, at the tips of the mitral leaflets, perpendicular to the transmitral inflow: (1) transmitral through-plane flow velocity (encoding velocity Venc = 180 cm/s, TE = 3.1 ms, TR = 7.6 ms, views per segment = 2; temporal resolution = 15 ms), and (2) longitudinal myocardial velocity (Venc = 15 cm/s or 20 cm/s, TE = 5 ms, TR = 9.5 ms, views per segment = 2; temporal resolution = 20 ms). For both sequences, the following parameters were used: flip angle = 20°, slice thickness = 8 mm, pixel spacing = 1.9 mm × 1.9 mm, matrix 256 × 128. To minimize background offsets and for the acquisition time to remain compatible with breath-holding, a 50% rectangular field-of-view was used [22]. Each dataset included dynamic modulus series (providing information about the variation in mitral valve orifice geometry during the cardiac cycle) and the associated velocity-encoded dynamic series, which were acquired during a complete cardiac cycle. These contours were then superimposed on velocity PC-CMR images for flow analysis.

### 2.3. CMR Analysis

#### 2.3.1. Quantitative Assessment of the LV Functions

All images were evaluated by two experienced observers, blinded to all clinical data. LVEDV and LV end-systolic volume (LVESV), LVEF and LV mass (LVM) were measured on short-axis cine-SSFP images. Epicardial and endocardial borders were traced semi-automatically at end-diastole and end-systole using specialized software (Syngo.Via VB20A_HF04, Argus, Siemens Medical Solutions). All volumes were indexed to body surface area. The LV sphericity index (LVSI) was calculated by dividing LVEDV by the volume of a sphere whose LV length (L) was measured at the end of diastole: LVSI = LVEDV/[π/6 × (L)^3^] [23].

LGE was assessed from short-axis images, using the 17-segments model [24] and was quantified using a signal intensity threshold of >5SD above a remote reference for normal myocardium, as recommended by international protocols [25], using dedicated software (cvi42, Circle Cardiovascular Imaging Inc., Calgary, CA, USA). 

Applying the previously described algorithms, three basic waveforms were obtained, which allowed measurements of the following parameters using dedicated software (cvi42, Circle Cardiovascular Imaging Inc., Calgary, CA, USA): transmitral early (E, in cm/s) and late (A, in cm/s) peak velocities and early (EQ, in mL/s) and late (AQ, in mL/s) peak flow-rates. Transmitral filling volume (FV), which was estimated as the area under the trans-mitral and flow–rate curve between the automatically detected beginning and end of the filling period; transmitral deceleration time (DT, ms) and isovolumic relaxation time (IVRT, ms) were also determined. Myocardial longitudinal early (E’, in cm/s) and late (A’, in cm/s) peak velocity on LV lateral wall. Of note, the same dataset used for transmitral flow analysis was used to extract aortic ejection flow–rate curves and to identify the end of ejection time, which was used for IVRT estimation.

#### 2.3.2. Quantitative Assessment LA Volume and Function

LA volumes (LAV) were measured at different moments of the cardiac cycle: maximum LA volume at LV end-systole, just before mitral valve opening (LAV_max_), pre-atrial contraction LA volume (LAV_preA_) at LV diastole immediately prior to LA contraction, and minimum LA volume at mitral valve closure and late LV diastole after LA contraction (LAV_min_). Each phase was visually determined and LA volumes were calculated using apical 4- and 2-chamber views. In the 4-chamber view, the LA border started from the medial side of the mitral annulus, included interatrial septum, posterior, and lateral LA walls and ended at the lateral mitral annulus. In the 2-chamber view, the analyzed LA border started from anterior mitral annulus and continued over the LA roof, the posterior wall, and the floor of the LA and ended at the inferior mitral annulus. LA length was the long-axis length of LA from each chamber. The LA endocardial border was manually delineated using both the 2-and 4-chamber cine images, although pulmonary veins confluence and LA appendage were excluded. The LAV was calculated using this formula: LA volume = (0.848 × area 4-chamber × area 2-chamber)/([length 4-chamber + length 2-chamber]/2) [6,26]. LASI was calculated using this formula: LA volume = maximum LA volume/(4π/3)(maximum LA length/2) [19]. In all patients, LAV was indexed to body surface. LA reservoir function was described by using the LA total emptying volume = LAV_max_ − LAV_min_ and the LA total emptying fraction (LATF) = (LAV_max_ − LAV_min_)/LAV_max_. The following parameters were measured to evaluate the LA conduit function: the LA passive emptying volume = LAV_max_ − LAV_preA_ and LA passive emptying fraction (LAEF) = (LAV_max_ − LAV_preA_) /LAV_max_.Parameters of atrial booster pump function included: the LA active emptying volume = LAV_preA_ − LAV_min_ and the LA active emptying fraction (LAAF) = (LAV_preA_ − LAV_min_)/LAV_preA_ [27]. 

CMR atrial strain was analysand processed using commercial cardiovascular postprocessing software QStrain (version 3.1.16.9, MedisMedical Imaging Systems, Leiden, The Netherlands). Afterwards, according to the LA’s phasic functions, specific LA strains were then evaluated: LA-ε_t_ for the LA reservoir function, LA-ε_p_ for the LA passive emptying function, and LA-ε_a_ for the LA booster pump function [12].

### 2.4. Follow-Up of Clinical Outcomes

The clinical follow-up was obtained by completing questionnaires either during hospital visits, by telephone, or both, aiming at delineating the occurrence of the clinical outcomes, which corresponded to the first event occurring in each patient among the following MACEs: death, non-fatal cardiac arrest, ventricular tachyarrhythmia, and heart failure requiring hospitalization defined accordingly to current international guidelines. Survival analysis was performed for the clinical outcomes. The median follow-up was 26 months and maximum follow-up reached 41 months.

### 2.5. Statistical Analysis

All data were tested for normality using the Kolmogorov–Smirnov test. Continuous data were presented as median (inter-quartile range [IQR]) and mean ±standard deviation (SD). Discrete data were reported as percentages and frequencies. Baseline clinical and CRM data were analyzed according to LAε tertiles, using a trend test for categorical variables and 1-way ANOVA with Bonferroni correction or its nonparametric equivalents (Kruskal–Wallis test) for continuous variables. The correlations between the LAV index, parameters LA function and diastolic function were measured using the Pearson’s or Spearman correlation test, and the predictors for the LAV index were assayed using a linear regression model. Hazard ratios (HR) for the prediction of events were calculated using Cox regression models. For each outcome, we considered all the significant variables in the univariate analysis and sought the best overall multivariate models for the composite endpoint, by stepwise-forward selection, with a probability to enter set at *p* < 0.05 and to remove the effect of regression at *p* < 0.05. Time to event outcomes were analyzed using the Kaplan–Meier method and compared using the log-rank test. Multivariate analysis was performed using a multiple logistical regression model for generation of hazard ratios (95% confidence interval). Cohen’s Kappa inter- and intra-observer coefficients were determined. Retrospective test power calculation and prospective sample size were estimated, with type I and type II variation according to sample size. Statistical significance was set as *p*-value < 0.05. Statistical analysis was performed using MedCalc (Version 19.1.7, MedCalc Software, Belgium).

## 3. Results

### 3.1. Baseline Characteristics

Overall, 212 patients with NIDCM (49 ± 14.2-year-old; 73.5% males) (Figure 1) and 106 healthy subjects who served as controls (49 ± 11.2-year-old; 70.7% males) were finally included in the study (Table 1). 

There were no significant differences between the two groups in terms of clinical characteristics. However, biomarkers of cardiac dysfunction and fibrosis were considerably higher in the diseased group (*p* < 0.001).

Regarding the reproducibility of CMR parameters, measurements were repeatedly performed on the same set of images. For LAV_max_, LAV_min_, LAV_preA_, LASI and E/E’ ratio, the intra- and inter-observer reproducibility was very good, with kappa coefficients of inter-observer agreement 0.90 for LAV_max_, 0.93 for LAV_min_, 0.87 for LAV_preA_, 0.94 for LASI and 0.92 for E/E’ ratio, while the intra-observer coefficients were 0.93 for LAV_max_, 0.95 for LAV_min_, 0.91 for LAV_preA_, 0.94 for LASI and 0.96 for E/E’ ratio.

### 3.2. Characterization of LV Systolic and Diastolic Functions

CMR measurements showed an impaired LV systolic function in those with NIDCM (Table 2), characterized by substantially increased LVEDV, LVESV and LVM, and decreased LVEF (*p* < 0.001). LAS and LVSI were also defective in the NIDCM group (*p* < 0.001). LV diastolic function and transmitral parameters were notably impaired in the NIDCM group, namely E, DT, E/E’ ratio (all *p* < 0.001). Moreover, CMR flow parameters of EQ (*p* < 0.001), AQ (*p* < 0.001), QE/QA ratio (*p* < 0.05), and EQ/LVEDV (*p* < 0.001) were also significantly impaired in the NIDCM group.

### 3.3. Characterization of LA Phasic Function and Geometry

CMR measurements of LA phasic function and geometry are presented in Table 3. LA volumes were significantly increased in NIDCM group (all *p* < 0.001). All three LA phasic functions and strain patterns were significantly impaired in the NIDCM group, namely for LA reservoir function: LATF (43.7% ± 8.5 vs. 58.9% ± 2.9 (*p*  <  0.001) and LA-ε_t_ (31.5% ± 2.4 vs. 39.8% ± 2.8, *p* < 0.001), for LA conduit function: LAPF (20.9% ± 7.2 vs. 29.1% ± 8.4, *p* < 0.001), LA-ε_p_ (16.0% ± 4.1 vs. 17.9% ± 2.7, *p* < 0.001) and for LA atrial booster function: LAAF (28.6% ± 9.6 vs. 41.3% ± 8.3, *p*  <  0.001), LA-ε_a_ (20.6% ± 2.5 vs. 29.2% ± 3.6, *p* < 0.001).

LASI was significantly increased in the NIDCM group compared to the healthy volunteers group (0.77 vs. 0.39, *p* < 0.001), while 57% of patients were positive for atrial replacement fibrosis represented by LA-LGE.

### 3.4. Correlations between LA Function, Geometry, and LV Function Parameters in Patients with NIDCM

The most relevant correlations between LA function, geometry and LV function are presented in Table 4. LA phasic function and strain parameters were inversely associated with LA geometry. Amongst them, LAV_min_ proved to have the most powerful correlations with LATF, LAPF, LAAF, LA-ε_t_ and LA-ε_a_ (*p* < 0.0001), while LAV_pre-A_ and LASI were correlated with LAPF and LA-ε_t_ (*p* < 0.0001).

Regarding LA function and LV conventional function parameters, apart from LVEF which was positively associated with all LA functional parameters, other LV function measurements were negatively correlated with the LA ones. LVEF had the strongest associations with LATF, LAAF, LA-ε_t_ and LA-ε_a_ (*p* < 0.0001). LVEDV and LVESV were negatively correlated with LATF, LAAF, LA-ε_t_ and LA-ε_a_ (*p* < 0.0001). LAS was also inversely associated with LATF, LAAF, LA-ε_t_ and LA-ε_a_ (*p* < 0.0001).

### 3.5. Univariate and Multivariate Analysis of LA Function and Geometry Parameters

Patients with NIDCM were followed-up for a median of 26 months. Of all, 7 experienced cardiac death, 14 ventricular tachyarrhythmias, and 19 heart failure hospitalization. Cox regression analyses were performed to evaluate the ability of LA parameters to predict the composite outcome (Table 5). 

In unadjusted analysis, LA volumes, LASI, LA-LGE and LA-ε_t_ were significantly correlated with increased risk for the outcome and persisted even after adjustments for specific clinical and CMR covariates. Furthermore, in unadjusted analysis, LASI had an HR for the composite outcome of 2.07 (*p* < 0.001). This decreased to 1.16 (*p* < 0.001) in multivariate analysis. LA-ε_t_ was associated with the increased composite outcome (HR 4.14, *p* < 0.0001). The HR decreased slightly to 3.81 when covariates were added to the analysis. The prediction ability of LA-LGE increased after the adjustment of covariates from HR of 3.83 (*p* < 0.001) to 4.52 (*p* = 0.0001). 

### 3.6. Time to Event Analysis and Incremental Predictive Ability of LA Function and Geometry Parameters

Kaplan–Meier analysis was used to test the ability of LA volumes, LASI and LA-ε_t_ for predicting the composite outcome. Hence, for LA geometry, Kaplan–Meier analysis demonstrated higher risk for composite outcomes with higher LA volumes (LAV_max_ > 54 mL/m^2^ [HR = 2.02; 95%CI (1.07–3.83), *p* < 0.03)], LAV_min_ > 31 mL/m^2^ [HR =1.66; 95%CI (1.08–3.15), *p* < 0.01], LAV_pre-A_ > 43 mL/m^2^ [HR = 2.11; 95%CI (1.11–3.99), *p* < 0.02]), and with LASI > 0.725, (HR = 2.68; 95%CI (1.42–5.07), *p* < 0.003) (Figure 2).

Patients were divided in three subgroups based on the LA function and according to LA-ε_t_ tertiles, to evaluate if the severity of LA phasic dysfunction impacts the prediction of the composite outcome. Hence, 42 patients had mild impairment of LA-ε_t_ > 35% (tertile 1), 58 patients had moderate impairment of LA-ε_t_ 30–35% (tertile 2) and 112 patients had severe impairment of the LA-ε_t_ < 30% (tertile 3). Kaplan–Meier analysis demonstrated that patients with LA-ε_t_ < 30% had a higher risk for the composite outcome. Additionally, during the follow-up period, the composite outcome occurred in 25.9% of patients with LA-ε_t_ < 30% (tertile 3), while occurrence in the other two categories was lower (Figure 3).

Stepwise Cox proportional-hazards models showed that these parameters had considerable predictive value for the composite outcome (Figure 4). Foremost, the addition of LASI to LA-ε_t_ significantly increased the predictive ability (Chi-square = 10.2, *p* < 0.001), while the further addition of LA-LGE to them enhanced it even more (Chi-Square = 15.8; *p* < 0.0001).

## 4. Discussion

In this prospective study, it was shown that a comprehensive appraisal of LA function and geometry using CMR provides valuable information in patients with NIDCM. The major findings of the current research comprise (1) LA function and geometry parameters were significantly impaired in those with NIDCM, when compared to the control group; (2) afflicted LA function and strain parameters were correlated with impaired LA geometry and LV dysfunction; (3) LA-ε_t_, LASI and LA-LGE were considerably better at predicting composite outcome than other LA measurements, being also independently correlated with it; (4) the severity of LA-ε_t_ was significantly associated with the outcome; (5) LA-ε_t_, LASI and LA-LGE added incremental predictive value for the outcome, beyond other parameters. Furthermore, this is the first study to assess the predictive ability of LA-ε_t_ by CMR in patients with NIDCM.

As a direct result of impaired LV, LA enlargement and dysfunction with defective LA phasic functions are frequent in cardiac conditions [5,11,28,29,30]. First and foremost, studies that analyzed the role of echocardiography in LA remodelling, phasic function and geometry have begun to emerge. They had shown that LA phasic function parameters were strongly associated with HF, impaired LV relaxation and compliance, atrial fibrillation and AHT [10,20,31,32,33]. LA conduit function was found to be defective even before LV hypertrophy and enlargement occurred [34], and to be directly associated with diastolic dysfunction, its severity [32] and progression [35]. Moreover, LA reservoir function was notably linked to cardiac dysfunction in diabetic [36] and cancer patients [37]. Additionally, the role of LA strain had been certified, being considerably linked to disease severity, HF hospitalization and mortality [38].

The role of CMR in evaluating these parameters has recently been endorsed by certain studies, but more work is required [11,27,31]. In the current study, all CMR measurements of LA dysfunction and altered geometry were notably flawed. Due to the lack of specific thresholds, these findings required validation by comparison with healthy individuals which served as a control group. Hence, CMR proved tremendous ability in characterizing LA malfunction and remodelling. Moreover, LA volumes were significantly increased in the diseased group, akin to other published data [2,3,14,15,32]. All LA phasic functions were considerably defective when compared with the control group, while corresponding LA strains were significantly impaired in patients with NIDCM, analogously to other published research [12]. Withal, in the diseased group, LASI was found to be considerably increased, whereas over half of patients had LA irreversible replacement fibrosis measured by LA-LGE, both being recently recognized as markers of LA remodelling [39,40,41].

Thereafter, the appositeness of LA function and geometry parameters was subsequently tested. The LA phasic function parameters were inversely related to LA volumes and geometry measurements, thus suggesting that LA dilation and remodelling are closely linked to defective LA function. Notwithstanding, LAV_min_, LAV_pre-A_ and LASI correlated the best both reservoirs (LATF and LA-ε_t_) and atrial booster functions (LAAF and LA-ε_a_). The relevancy of LA volumes and LASI had been endorsed by recently published data, especially regarding LAV_min_ and LASI [19,42]. LVEDV and LVESV were inversely associated with LA phasic functions, remarkably with LA reservoir and atrial booster functions. Furthermore, LVEF was positively linked to these functional measurements, albeit its progressive worsening was closely related to LA dysfunction. These findings are comparable to other published studies, which had also found substantial associations between impaired LA phasic functions and LV dysfunction [11,24,25,30].

Furthermore, relevant LA parameters were selected using the regression coefficients and comprised LA volumes, LA-ε_t_, LASI and LA-LGE, all providing significant predictive ability. LA volumes were independently associated with composite outcome, while Kaplan–Meier analysis showed significant predictive ability. Similarly, other published studies endorsed the importance of CMR-determined LA volumes as independent outcome predictors [42,43,44]. However, in contrast with other reported data, this is the first study to suggest that LA volume parameters were outranked by LA geometry and fibrosis measurements. These findings put forward that LA dysfunction has other determinants, besides LA dilation. Moreover, LASI proved an outstanding prognostic ability for the composite outcomes, even after the adjustment for covariates (model 4). Additionally, Kaplan–Meier analysis confirmed the predictive capacity of LASI > 0.725. This marker has been recently described as an indicator of LA remodelling, in both CMR and echocardiography studies, being able to predict atrial fibrillation’s recurrence and HF hospitalization [19,34,39]. To our knowledge, this is the first study to evaluate the ability of LASI to predict cardiovascular mortality.

Moreover, our study is the first to evaluate the prognostic role of LA-ε_t_ in patients with NIDCM. At first, the prognostication capacity of LA-ε_t_ was tested and showed a remarkable predictive ability for outcome. Thereafter, to evaluate the relationship between the severity of LA-ε_t_ and composite outcome, specific LA-ε_t_ tertiles were created and patients were segregated accordingly. Time to event analysis showed significant predictive ability, especially for those with severe impairment of LA-ε_t_ (third tertile; LA-ε_t_ < 30%). The predictive role of LA strain had been previously shown in both echocardiography and CMR studies, in other cardiovascular diseases [12,31,38,40]. To date, this is the first study to prove the predictive ability of LA-ε_t_ in NIDCM.

Lastly, using Cox regression models, the incremental predictive value of these parameters was evaluated. Hence, the embedment of LASI to LA-ε_t_ provided a significant incremental value, while the addition of LA-LGE improved even more the prediction of composite outcome. To our knowledge, this is the first study to evaluate the incremental predictive value of combined LA function, geometry and fibrosis parameters using CMR.

As for the study’s limitations, firstly, this is a single-center study. Secondly, phase-contrast volume CMR is still a novel method to assess the LA and several technical errors might have occurred. Thirdly, specific thresholds for LA-ε_t_ are lacking and reference intervals still require large cohort populational studies. Finally, patients did not benefit from cardiac resynchronization therapy with a defibrillator which might have positively influenced their prognosis.

## 5. Conclusions

In patients with NIDCM, LA function and geometry are substantially impaired and considerably associated with LV dysfunction. Parameters of LA function (LA-ε_t_), geometry (LASI) and myocardial fibrosis (LA-LGE) are independent predictors for HF hospitalization all-cause mortality. Furthermore, they provide incremental prognostication value beyond age, gender, LV conventional systolic and diastolic parameters, and LV geometry and strain.

## Figures and Tables

**Figure 1 biomedicines-09-01653-f001:**
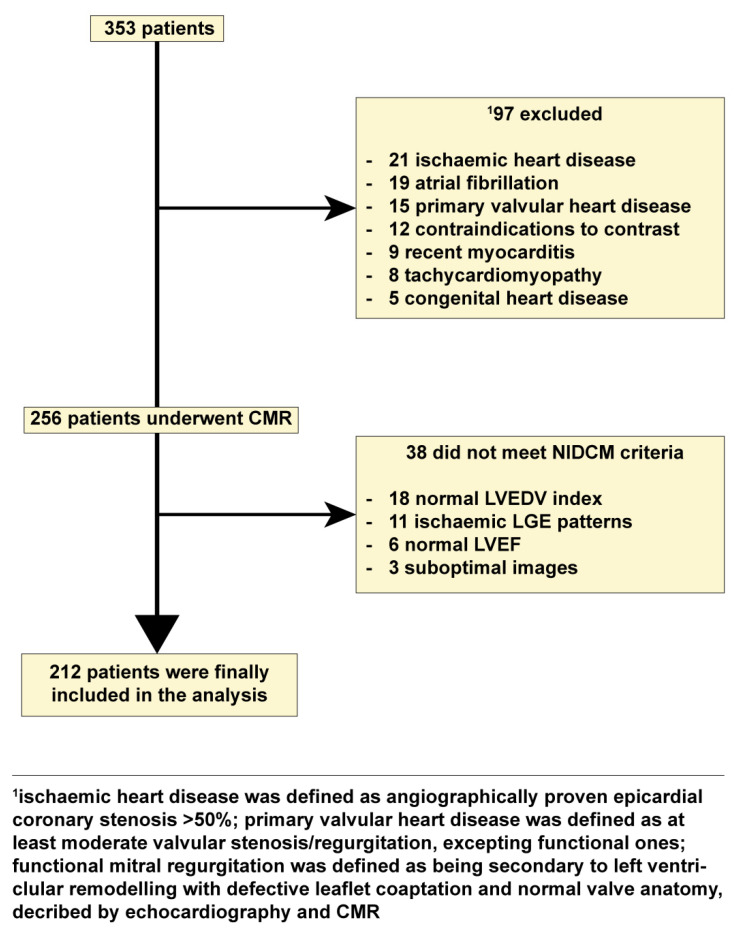
Flow chart detailing the study cohort. Abbreviations: CMR, cardiac magnetic resonance imaging; LGE, late gadolinium enhancement; NIDCM, nonischemic dilated cardiomyopathy.

**Figure 2 biomedicines-09-01653-f002:**
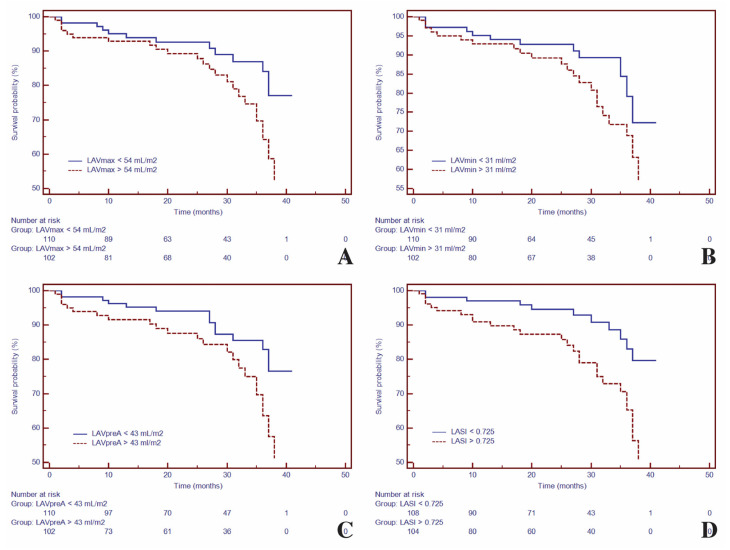
Kaplan–Meier curves for time to event analysis of: LAV_max_ (**A**), LAV_min_ (**B**), LAV_pre-A_ (**C**) and LASI (**D**). Abbreviations: LAV_max_, maximum left atrial volume; LAV_min_, minimum left atrial volume; LAV_pre-A_, pre-atrial contraction left atrial volume; LASI, left atrial sphericity index.

**Figure 3 biomedicines-09-01653-f003:**
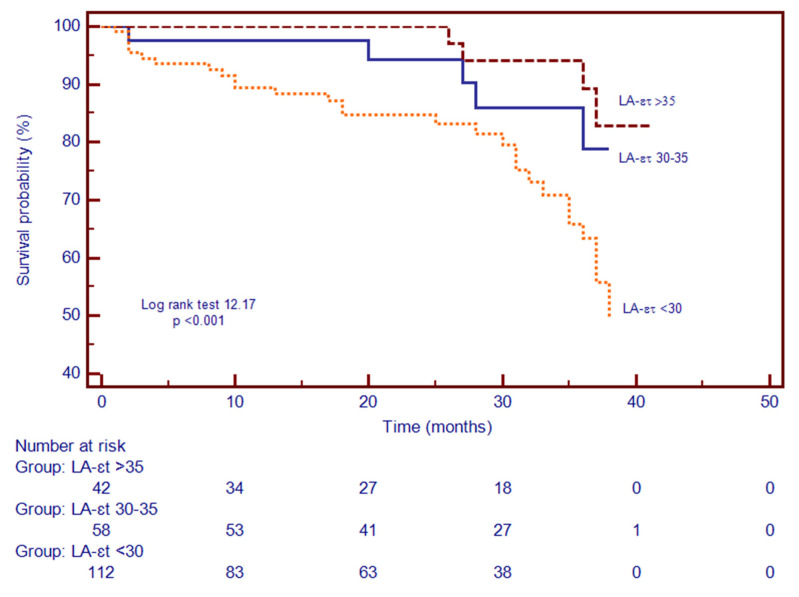
Kaplan–Meier curves for time to event analysis of LA-ε_t_. Abbreviations: LA-ε_t_, left atrial total strain.

**Figure 4 biomedicines-09-01653-f004:**
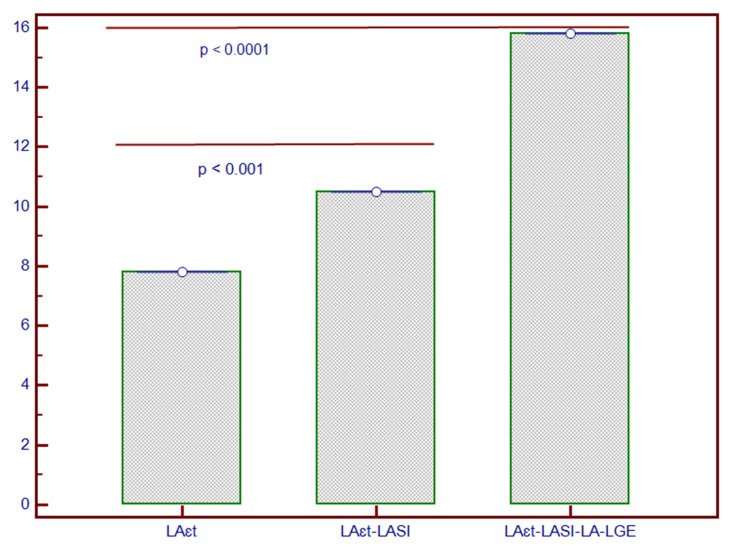
Incremental predictive value of LASI added to LA-ε_t_ and to LA-LGE for composite outcome in patients with NIDCM. * The y-axis represents the Chi-square values of the stepwise Cox proportional-hazard models. Abbreviations: LA-ε_t_, left atrial total strain; LA-LGE, left atrial late gadolinium enhancement; LASI, left atrial sphericity index; NIDCM, non-ischaemic dilated cardiomyopathy.

**Table 1 biomedicines-09-01653-t001:** Baseline characteristics of patients in study.

	NIDCM All Patients *n* = 212	Healthy Volunteers *n* = 106	*p*-Value
Clinical characteristics			
- Age, mean (SD), years	49 (14.2)	49 (11.8)	NS
- Male gender, *n* (%)	156 (73.5)	75 (70.2)	NS
- Body-mass index, kg/m^2^	27.4 (4.7)	27.8 (4.8)	NS
- Heart rate, mean (SD), bpm	73 (12.8)	73 (15.8)	NS
- Systolic blood pressure, mean (SD), mmHg	134 (19.1)	122 (17.2)	
- Hypertension, *n* (%)	51 (24.0)	20 (18.9)	<0.01
- Diabetes mellitus, *n* (%)	40 (18.8)	15 (14.1)	<0.01
- Dyslipidemia, *n* (%)	96 (45.2)	49 (46.2)	NS
- Smoking, *n* (%)	64 (30.2)	37 (34.9)	NS
Biomarker levels			
- NT-proBNP, median (IQR), pg/mL	2679 (378–11,960)	223 (60–391)	<0.001
- eGFR, mean (SD), mL/min/1.73 m^2^	86.4 (20.1)	87.4 (16.8)	NS
- PICP, median (IQR), ng/mL	1.45 (0.42–9.2)	0.47 (0.31–5.1)	<0.001
- PIIINP, median (IQR), ng/mL	15.6 (2.7–79.1)	8.2 (2.1–23.5)	<0.001
- Galectin-3, median (IQR), ng/mL	13.9 (2.2–21.6)	4.8 (1.2–12.8)	<0.001

Abbreviations: *n*, number of patients; SD, standard deviation; IQR, interquartile range; NS, not significant; NIDCM, non-ischemic dilated cardiomyopathy; NT-proBNP, N-terminal pro-Brain Natriuretic Peptide; eGFR, estimated glomerular filtration rate; PICP, procollagen type I C-terminal propeptide; PIIINP, procollagen type III N-terminal propeptide; Data are reported as mean (standard deviation) or median (IQR) or *n* (%).

**Table 2 biomedicines-09-01653-t002:** Cardiac magnetic resonance imaging indices of systolic and diastolic function.

Variables	NIDCM All Patients *n* = 212	Healthy Volunteers *n* = 106	*p*-Value
LV and RV systolic parameters			
- LVEDV index, mL/m^2^	131.5 (36.5)	63.1 (18.6)	<0.001
- LVESV index, mL/m^2^	87.1 (36.2)	21.6 (7.4)	<0.001
- LVM index, g/m^2^	86.1 (20.1)	58.4 (14.1)	<.001
- LVEF, %	35.5 (10.1)	65.7 (5.6)	<0.001
- LAS, %	−9.5 (5.4)	−20.0 (2.7)	<0.001
- LVSI	0.40 (0.12)	0.18 (0.05)	<0.001
- LV-LGE mass, g	34.4 (13.1)	-	NA
- LV-LGE mass/LVM, %	18.1 (7.7)	-	NA
- RVEDV index, mL/m^2^	57.1 (24.0)	59.0 (19.3)	NS
- RVESV index, mL/m^2^	31.2 (17.6)	22.5 (8.1)	<0.001
- RVEF, %	46.8 (9.5)	61.8 (5.7)	<0.001
- TAPSE, mm	16.6 (5.1)	21.2 (2.9)	<0.001
LV diastolic parameters			
- E, cm/s	52.1 (12.2)	75.5 (9.3)	<0.001
- A, cm/s	34.5 (9.4)	47.2 (6.9)	NS
- DT, ms	210 (80.3)	182 (58.5)	<0.001
- IVRT, ms	60 (18.7)	88 (18.1)	NS
- E’, cm/s	6.2 (2.3)	5.8 (2.2)	<0.01
- A’, cm/s	4.7 (1.2)	3.4 (1.3)	NS
- E/A ratio	1.46 (0.93)	1.85 (0.73)	<0.05
- E/E’ratio	9.5 (2.8)	6.5 (1.5)	<0.001
- EQ, mL/s	243 (112.8)	344 (90.7)	<0.001
- AQ, mL/s	213 (122.5)	205 (75.2)	NS
- EQ/LVEDV, s^−1^	1.99 (0.61)	3.83 (1.42)	<0.001
- EQ /FV, s^−1^	4.01 (1.2)	4.9 (0.82)	NS
- FV, mL	61.9 (21.4)	71.7 (20.1)	NS

Abbreviations: *n*, number of patients; SD, standard deviation; IQR, interquartile range; NA, not applicable; NS, not significant; NIDCM, non-ischemic dilated cardiomyopathy; FV, mitral filling volume; LAS, left ventricular longitudinal-axis strain; LV-LGE, left ventricular late gadolinium enhancement; LVEDV, left ventricular end-diastolic volume; LVESV, left ventricular end-systolic volume; LVM, left ventricular mass; LVEF, left ventricular ejection fraction; LVSI, left ventricular sphericity index; A, late peak mitral flow velocity; A’, myocardial longitudinal late diastolic peak myocardial velocity; E, early peak mitral flow velocity; E’, myocardial longitudinal early diastolic peak myocardial velocity; AQ, late peak mitral flow-rate; EQ, early peak mitral flow-rate; DT, early diastolic filling deceleration time; IVRT, isovolumic relaxation time, TAPSE, tricuspid annular plane systolic excursion; RVEDV, right ventricular end-diastolic volume; RVESV, right ventricular end-systolic volume; RVEF, right ventricular ejection fraction. Data are reported as mean (standard deviation) or median (IQR) or *n* (%).

**Table 3 biomedicines-09-01653-t003:** Comparison between left atrial function and geometry parameters between NIDCM and healthy volunteers.

Variables	NIDCM All Patients *n* = 212	Healthy Volunteers *n* = 106	*p*-Value
LA volumes indexed			
- LAV_max_ index, mL/m^2^	56.1 (21.7)	34.0 (6.2)	<0.001
- LAV_min_ index, mL/m^2^	32.4 (9.5)	14.0 (2.7)	<0.001
- LAV_pre-A_ index, mL/m^2^	45.3 (11.1)	24.1 (4.7)	<0.001
Reservoir function			
- LATF, %	43.7 (8.5)	58.9 (2.9)	<0.001
- LA-ε_t_, (%)	31.5 (2.4)	39.8 (2.8)	<0.001
Conduit function			
- LAPF, %	20.9 (7.2)	29.1 (8.4)	<0.001
- LA-ε_p_, (%)	16.0 (4.1)	17.9 (2.7)	<0.001
Atrial booster function			
- LAAF, %	28.6 (9.6)	41.3 (8.3)	<0.001
- LA-ε_a_, (%)	20.6 (2.5)	29.2 (3.6)	<0.001
LA geometry			
- LASI	0.77 (0.23)	0.39 (0.06)	<0.001
- LA-LGE +, *n* (%)	121 (57)	-	NA
Severe mitral regurgitation, *n* (%)	60 (28.3)	-	NA

Abbreviations: LA-ε_a_, left atrial active strain; LA-ε_p_, left atrial passive strain; LA-ε_t_, left atrial total strain; LAAF, left atrial active emptying function; LAPF, left atrial passive emptying function; LASI, left atrial sphericity index; LATF, left atrial total emptying function; LAV_max_, maximum left atrial volume; LAV_min_, minimum left atrial volume; LAV_pre-A_, pre-atrial contraction left atrial volume; LA-LGE, left atrial late gadolinium enhancement; NA, not applicable; NIDCM, non-ischemic dilated cardiomyopathy.

**Table 4 biomedicines-09-01653-t004:** Correlations between LA function and parameters of LA geometry and LV function in the NIDCM group.

	LA Phasic Function Parameters					
	LATF	LAPF	LAAF	LA-ε_t_	LA-ε_p_	LA-ε_a_
**LA geometry parameters**						
LAV_max_ index, mL/m^2^	−0.489 (<0.0001)	−0.201 (0.003)	−0.464 (<0.0001)	−0.459 (<0.0001)	−0.222 (0.001)	−0.394 (<0.0001)
LAV_min_ index, mL/m^2^	−0.817 (<0.0001)	−0.419 (<0.0001)	−0.713 (<0.0001)	−0.663 (<0.0001)	−0.273 (<0.001)	−0.597 (<0.0001)
LAV_pre-A_ index, mL/m^2^	−0.682 (<0.0001)	−0.519 (<0.0001)	−0.484 (<0.0001)	−0.641 (<0.0001)	−0.208 (0.002)	−0.602 (<0.0001)
LASI	−0.587 (<0.0001)	−0.263 (<0.001)	−0.546 (<0.0001)	−0.635 (<0.0001)	−0.277 (<0.001)	−0.565 (<0.0001)
**LV conventional function**						
LVEDV index, mL/m^2^	−0.583 (<0.0001)	−0.336 (<0.0001)	−0.500 (<0.0001)	−0.655 (<0.0001)	−0.199 (0.004)	−0.627 (<0.0001)
LVESV index, mL/m^2^	−0.593 (<0.0001)	−0.343 (<0.0001)	−0.506 (<0.0001)	−0.661 (<0.0001)	−0.201 (0.003)	−0.634 (<0.0001)
LVEF, %	0.673 (<0.0001)	0.386 (<0.0001)	0.576 (<0.0001)	0.765 (<0.0001)	0.223 (0.001)	0.736 (<0.0001)
LVSI	−0.150 (0.02)	−0.170 (0.01)	−0.058 (0.39)	−0.156 (0.02)	−0.073 (0.28)	−0.114 (0.09)
LAS, %	−0.618 (<0.0001)	−0.336 (<0.0001)	−0.537 (<0.0001)	−0.628 (<0.0001)	−0.175 (0.018)	−0.619 (<0.0001)
E/E’ ratio	−0.400 (<0.0001)	−0.260 (<0.001)	−0.313 (<0.0001)	−0.387 (<0.0001)	−0.063 (0.26)	−0.399 (<0.0001)

Abbreviations: NIDCM, non-ischemic dilated cardiomyopathy; LA-ε_a_, left atrial active strain; LA-ε_p_, left atrial passive strain; LA-ε_t_, left atrial total strain; LA, left atrium; LAAF, left atrial active emptying function; LAPF, left atrial passive emptying function; LAS, left ventricular long-axis strain; LASI, left atrial sphericity index; LATF, left atrial total emptying function; LAV_max_, maximum left atrial volume; LAV_min_, minimum left atrial volume_;_ LAV_pre-A_, pre-atrial contraction left atrial volume; LV, left ventricle; LVEDV, left ventricular end-diastolic volume; LVEF, left ventricular ejection fraction; LVESV, left ventricular end-systolic volume; LAS, left ventricular longitudinal-axis strain; LVSI, left ventricular sphericity index; E, early peak mitral flow velocity; E’, myocardial longitudinal early diastolic peak myocardial velocity. Table data are the coefficients of correlations.

**Table 5 biomedicines-09-01653-t005:** Univariable and Multivariable Cox Regression Analyses: Association of left atrial functional indices with the composite end point.

	Univariate HR (95%CI)	Multivariate			
		Model 1	Model 2	Model 3	Model 4
LAV_max_ index	1.02 (1.01–1.03) *p* < 0.01	1.01 (1.00–1.02) *p* < 0.01	1.02 (1.00–1.03) *p* < 0.01	1.01 (1.01–1.03) *p* < 0.01	1.02 (1.00–1.05) *p* < 0.001
LAV_min_ index	1.03 (1.00–1.04) *p* < 0.01	1.04 (1.01–1.07) *p* <0.01	1.05 (1.02–1.08) *p* <0.001	1.03 (1.00–1.07) *p* < 0.01	1.04 (1.02–1.07) *p* < 0.001
LAV_pre-A_ index	1.03 (1.01–1.05) *p* < 0.01	1.06 (1.01–1.06) *p* < 0.01	1.07 (1.02–1.12) *p* < 0.001	1.05 (1.01–1.07) *p* < 0.001	1.06 (1.04–1.11) *p* < 0.001
LASI	2.07 (1.37–4.52) *p* < 0.001	1.83 (1.22–4.10) *p* < 0.001	1.74 (1.02–3.77) *p* < 0.001	1.76 (1.14–3.18) *p* < 0.001	1.16 (1.03–1.32) *p* < 0.01
LA-LGE	3.83 (1.90–7.71) *p* < 0.001	4.12 (2.03–8.31) *p* = 0.0001	4.03 (1.95–8.31) *p* < 0.001	4.04 (1.95–8.33) *p* < 0.001	4.52 (2.03–10.06) *p* = 0.0001
LA-ε_t_	4.14 (2.09–8.18) *p* < 0.0001	4.46 (2.23–8.91) *p* < 0.0001	4.38 (2.18–8.77) *p* < 0.0001	4.61 (2.27–9.36) *p* < 0.0001	3.81 (1.78–8.12) *p* < 0.001

Abbreviations: EQ, trans-mitral early peak flow-rate; LAV_max_, maximum left atrial volume; LAV_min_, minimum left atrial volume; LAV_pre-A,_ pre-atrial contraction left atrial volume; LA-LGE, left atrial late gadolinium enhancement; LASI, left atrial sphericity index; LATF, left atrial total emptying fraction; LA-ε_t_, left atrial total strain; LVSI, left ventricular sphericity index; LVEDV, left ventricular end-diastolic volume. Adjustment models: age, gender with the addition of significant parameters of univariable analysis. Data are hazard ratio (95% CI). Model 1 = adjusted for age + gender + ratio E/E’ +EQ/LVEDV Model 2 = Model 1 + mitral regurgitation severity Model 3 = Model 2 + LATF Model 4 = Model 3 + LAS + LVSI.

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
