# Peer review of "Left Atrial Geometry and Phasic Function Determined by Cardiac Magnetic Resonance Are Independent Predictors for Outcome in Non-Ischaemic Dilated Cardiomyopathy"

_biomedicines, 2021, doi:10.3390/biomedicines9111653_

Round 1

Reviewer 1 Report

In this study, the authors investigated the association between LA function/geometry and clinical outcomes in patients with DCM by comparing with control group without DCM. They found that LA function/geometry were substantially impaired and associated with LV dysfunction in patients with DCM. LA function, LA geometry, and myocardial fibrosis were independent predictors of heart failure hospitalization and all-cause mortality even after the adjustment for other potential confounders.

  1. Could the authors explain the reason of inclusion criteria?

  1. Could the authors explain a little bit more detail of the protocol to include control group.

  1. Could the authors explain the reason of why several variables were used for the adjustment?

  1. Could the authors add the definition of y-axis in Figure 4?

Author Response

Dear Editor,

We wish to thank the reviewers for their kind and helpful answers regarding our research paper. We are truly grateful for sharing their point of views and highlighting the shortcomings of our article and, thus, helping us to make the best of it.

Therefore, we tried to answer their highlights as good as we could:

Recenzor 1:

In this study, the authors investigated the association between LA function/geometry and clinical outcomes in patients with DCM by comparing with control group without DCM. They found that LA function/geometry were substantially impaired and associated with LV dysfunction in patients with DCM. LA function, LA geometry, and myocardial fibrosis were independent predictors of heart failure hospitalization and all-cause mortality even after the adjustment for other potential confounders.

Could the authors explain the reason of inclusion criteria?

We wish to thank the reviewer for this important remark. Several reasons made us use those inclusion criteria: firstly, we used the definition of dilated cardiomyopathy according to the European Society of Cardiology (particularly The diagnostic work up of genetic and inflammatory dilated cardiomyopathy); secondly the purpose of the study was to assess patients who were evaluated using cardiac magnetic resonance imaging (CMR), which is also the gold-standard imaging tool for evaluating cardiac function, overcoming the flaws of standard echocardiography. 

Could the authors explain a little bit more detail of the protocol to include control group.

We wish to appreciate the interest of the reviewer in unraveling this information gap. We chose to include 106 subjects who were overt of cardiovascular and respiratory disease (without chest pain, shortness of breath, palpitations, syncope, or cough) and had normal electrocardiograms. Also, echocardiography revealed no significant cardiac disease. Moreover, we stated that they had negative CMR; by using this affirmation, we meant that we included only subjects without any structural or functional heart disease which might have been discovered using CMR. On the other hand, we sought to use an identical evaluation protocol for this group of subjects as for the diseased one in order to rule out any possible bias.

Could the authors explain the reason of why several variables were used for the adjustment? 

This is another interesting remark that the reviewer pointed out. In our study we aimed to evaluate the predictive ability of LA parameters and, in order to assess their independency in terms of predicting the composite outcome, we constructed specific models of adjustment based on general and particular characteristic of patients. Moreover, when we sought to prove that their predictive ability is significant, we tried to cover all grounds. Thus, model 1 (age, gender, and LV diastolic dysfunction) was used to assess if the well-known codependency between LA parameters LV diastolic dysfunction is responsible for the predictive ability of LA parameters, or there is something more. Additionally, we initially tested for age and gender and excluded dependency, however we did not consider that it would have been necessary to construct a model only based on these two criteria. Furthermore, the purpose of Model 2 (Model 1 + the severity of mitral regurgitation) was to evaluate if mitral regurgitation and its severity plays a considerable role in the predictive ability of LA parameters and, as we have shown, the adjustment was still significant. As for Model 3 (Model 3 + LATF), the main goal for creating this model was to see if all the LA function parameters are independent from LA total emptying fraction or they have statistical significance only due to LATF and, thus, they are a surrogate for this measurement without any supplemental predictive ability. Last but not least, as expected, Model 4 (Model 3 + LV systolic function parameters) was created to provide evidence that LA parameters are also independent from LV systolic function parameters and, thus truly proving their independent predictive ability for outcome in this category of patients.

Could the authors add the definition of y-axis in Figure 4?

We wish to thank the reviewer for observing this gap. The y-axis of Figure 4 represents the value of chi-square for the stepwise Cox proportional-hazards models.

Reviewer 2 Report

I had the pleasure to review “Left atrial geometry and phasic function determined by cardiac magnetic resonance are independent predictors for outcome in nonischemic dilated cardiomyopathy by Bianca Olivia Cojan-Minzat et al. The authors compared the left atrial function determined by cardiac MRI between 212 patients with NIDCM and 106 controls. Furthermore, they performed correlation of LA function with long-term outcome and found especially LA total strain and LA-LGE to be associated with more MACE events within the NIDCM group.

In my opinion, this is a good manuscript of a well-performed study. However, I have the following minor comments:

  • The listed exclusion criteria at lines 82-85 do not correlate with the listed exclusion criteria at the study flowchart.
  • I do not understand if you incorporated LV measurements, such as LV function, into the multiple regression analysis. Please state if yes or no.
  • I do not understand the sentence “There were no significant differences between the two groups in terms of clinical characteristics or serum biomarker levels” (lines 218-219): The next sentence is a contradiction to the previous one.

Author Response

Dear Editor,

We wish to thank the reviewers for their kind and helpful answers regarding our research paper. We are truly grateful for sharing their point of views and highlighting the shortcomings of our article and, thus, helping us to make the best of it.

Therefore, we tried to answer their highlights as good as we could:

Recenzor 2:

I had the pleasure to review “Left atrial geometry and phasic function determined by cardiac magnetic resonance are independent predictors for outcome in nonischemic dilated cardiomyopathy by Bianca Olivia Cojan-Minzat et al. The authors compared the left atrial function determined by cardiac MRI between 212 patients with NIDCM and 106 controls. Furthermore, they performed correlation of LA function with long-term outcome and found especially LA total strain and LA-LGE to be associated with more MACE events within the NIDCM group.

In my opinion, this is a good manuscript of a well-performed study. However, I have the following minor comments:

The listed exclusion criteria at lines 82-85 do not correlate with the listed exclusion criteria at the study flowchart.

We thank the reviewer for observing this important flaw. The correct exclusion criteria from the flow-chart are the correct ones and, thus, we have made the proper corrections to the manuscript and added them to the text as well.

I do not understand if you incorporated LV measurements, such as LV function, into the multiple regression analysis. Please state if yes or no.

We wish to appreciate the reviewer for his interest in this matter. As you might notice in Table 5, Model 4 was constructed, besides the previously used measurements, using parameters of LV function such as LV long-axis strain (LAS) and LV sphericity index (LVSI). Therefore, the answer is yes.

I do not understand the sentence “There were no significant differences between the two groups in terms of clinical characteristics or serum biomarker levels” (lines 218-219): The next sentence is a contradiction to the previous one.

We wish to thank the reviewer for noticing this important mistake in our manuscript. We have corrected this gap accordingly: there were significant differences between the two groups in terms of serum biomarkers